Proceedings of the 7th Symposium on Advances in Approximate Bayesian Inference, 2025 1–24

# From predictions to confidence intervals: an empirical study of conformal prediction methods for in-context learning

**Zhe Huang**[*][†]
*PSL Research University, France*

**Simone Rossi**[*]
*EURECOM, France*

**Rui Yuan**
*Stellantis, France*

**Thomas Hannagan**
*Stellantis, France*

## Abstract

Transformers have become a standard architecture in machine learning, demonstrating strong in-context learning (ICL) abilities that allow them to learn from the prompt at inference time. However, uncertainty quantification for ICL remains an open challenge, particularly in noisy regression tasks. This paper investigates whether ICL can be leveraged for distribution-free uncertainty estimation, proposing a method based on conformal prediction to construct prediction intervals with guaranteed coverage. While traditional conformal methods are computationally expensive due to repeated model fitting, we exploit ICL to efficiently generate confidence intervals in a single forward pass. Our empirical analysis compares this approach against ridge regression-based conformal methods, showing that conformal prediction with in-context learning (*CP with ICL*) achieves robust and scalable uncertainty estimates. Additionally, we evaluate its performance under distribution shifts and establish scaling laws to guide model training. These findings bridge ICL and conformal prediction, providing a theoretically grounded and new framework for uncertainty quantification in transformer-based models.

## 1. Introduction

Transformers have become the standard building block in many machine learning domains, including natural language processing, computer vision, and reinforcement learning (Vaswani et al., 2017). Being expressive and readily parallelizable, these models achieve state-of-the-art performance on a wide range of tasks and are today the most widespread form of large language models (LLMs). Remarkably, transformers have also been shown to have good *in-context* abilities, i.e., they can follow examples given in the prompt to generate text that is consistent while seemingly learning from the prompt itself. This phenomenon, known as in-context learning (ICL) (Brown et al., 2020), has been tentatively explained in terms of meta-learning (Finn et al., 2017; Min et al., 2022), optimization (Von Oswald et al., 2023) and Bayesian reasoning (Falck et al., 2024; Ye et al., 2024). In particular, Von Oswald et al. (2023) demonstrated that self-attention layers can approximate gradient descent with a given learning rate (Rossi et al., 2024) by constructing task-specific updates to token representations.

---

[*] Equal contribution
[†] Work done during an internship at Stellantis.

In this paper, we explore the opportunities offered by in-context learning in relation to uncertainty quantification. We propose a novel method to estimate the uncertainty of transformers by leveraging their in-context learning capabilities. In particular, we attempt to answer the following research questions:

*Can we use the in-context learning capabilities of transformers to obtain robust and scalable uncertainty quantification in a noisy linear regression task? How does our method compare to existing uncertainty quantification methods?*

For the first question, we propose a method based on conformal prediction (Vovk et al., 2005; Shafer and Vovk, 2008), a framework for constructing prediction intervals that are guaranteed to have a certain coverage probability. Conformal prediction exhibits good theoretical properties and, being distribution-free, it does not require any assumptions on the distribution of data or model parameters. However, in its original form, conformal prediction is computationally greedy because it requires fitting a model multiple times with different data before confidence intervals can be produced. By exploiting in-context learning in transformers, we show that this computational issue can be avoided, leading to robust and scalable uncertainty quantification. We then leverage recent advances in mechanistic interpretations of in-context learning (Von Oswald et al., 2023) to build exact oracle conformal predictors, which we can compare to the performance of the proposed method. This allows us to assess the quality of our uncertainty estimates.

**Related work.** One of the motivations for scalable uncertainty quantification is to apply uncertainty quantification to LLMs. Uncertainty quantification in LLMs is an active area of research (see, e.g., Yin et al., 2023; Shorinwa et al., 2024). One possible approach to achieve it is to leverage the Bayesian inference machinery and recent advances in Bayesian deep learning to explicitly model the uncertainty of the model parameters (Neal, 1994; Blundell et al., 2015; Gal, 2016; Papamarkou et al., 2024). For example, Yang et al. (2023) propose a method to estimate the uncertainty of LLMs by combining the Low-Rank Adaptation (LoRA) methodology with a Bayesian approach based on the Laplace approximation (MacKay, 1998; Kristiadi et al., 2020). Alternatively, one can employ post-hoc calibration methods to estimate the uncertainty of LLMs (Guo et al., 2017; Lin et al., 2024). Closest to our work is the method by Ling et al. (2024), which proposes an uncertainty decomposition method for LLMs for in-context learning. In contrast, our approach leverages the ICL to perform uncertainty quantification through conformal prediction. Additionally, we focus on a simplified setting (noisy linear regression with linear self-attention) to provide a clear comparison with exact methods.

## 2. In-context learning and conformal prediction

In this paper we study the combination of distribution-free conformal prediction with in-context learning capabilities of transformer-based models. In this section, we start by providing a brief overview of the in-context learning setting and then we focus on the conformal prediction framework, which we will use to quantify the uncertainty of the predictions made by the model.

**Notation.** We follow a standard notation for vectors ($\boldsymbol{a}$), matrices ($\boldsymbol{A}$), and scalars ($a$). Given a vector $\boldsymbol{a}$, we denote its $i$-th element as $\boldsymbol{a}_{[i]}$. Given a matrix $\boldsymbol{A}$, we denote its $i$-th row and $j$-th column as $\boldsymbol{A}_{[i,j]}$.

## 2.1. Learning tasks in-context with transformer models

We are interested in analyzing the in-context learning setting for regression problems (Garg et al., 2022). The context for a task $\tau$ is a set of $n$ input features $\boldsymbol{x}_i \in \mathbb{R}^d$ and the corresponding noisy labels $y_i = f^{(\tau)}(\boldsymbol{x}_i) + \epsilon_i$, where $f^{(\tau)} : \mathbb{R}^d \to \mathbb{R}$ is the true function and $\varepsilon_i$ is the noise term. We denote this context as $\mathcal{D}_n^{(\tau)} = \{(\boldsymbol{x}_1, y_1), \ldots, (\boldsymbol{x}_n, y_n)\}$. In the ICL setting, given this context and a new input $\boldsymbol{x}_{n+1}$, we aim to predict the corresponding label $y_{n+1}$ with relatively good accuracy. Unlike classic learning methods, for which given task $\tau$ we need to train on the context $\mathcal{D}_n^{(\tau)}$ to estimate the function $f^{(\tau)}$ and make predictions, in ICL, learning and inference happen as a single forward pass through a pre-trained model. As we will see in a bit, this becomes a critical component of our methodology.

**Linear transformer models.** In this paper, we restrict our focus to transformer models (Vaswani et al., 2017) with linear-self attention layers (Von Oswald et al., 2023). Linear self-attention layers are a special case of self-attention layers where the attention weights are computed as a linear combination of the input features. Specifically, $\boldsymbol{f}_{\text{LSA}}(\boldsymbol{\theta}, \cdot) : \mathbb{R}^{(d+1) \times (n+1)} \to \mathbb{R}^{(d+1) \times (n+1)}$ is a linear self-attention layer parameterized by $\boldsymbol{\theta} = \{\boldsymbol{W}_K, \boldsymbol{W}_Q, \boldsymbol{W}_V, \boldsymbol{W}_O\}$, where $\boldsymbol{W}_K, \boldsymbol{W}_Q, \boldsymbol{W}_V, \boldsymbol{W}_O \in \mathbb{R}^{(d+1) \times (d+1)}$ are the key, query, value, and output weight matrices, respectively. Given a sequence of tokens $\boldsymbol{E} = [\boldsymbol{e}_1, \ldots, \boldsymbol{e}_{n+1}] \in \mathbb{R}^{(d+1) \times (n+1)}$, the output of the linear self-attention layer is computed as

$$\boldsymbol{f}_{\text{LSA}}(\boldsymbol{\theta}, \boldsymbol{E}) = \boldsymbol{E} + \boldsymbol{W}_O \boldsymbol{W}_V \boldsymbol{E} \left( \frac{(\boldsymbol{W}_K \boldsymbol{E})^\top \boldsymbol{W}_Q \boldsymbol{E}}{\sqrt{d}} \right) \tag{1}$$

This parameterization differs from the standard transformer model, where the attention weights are computed as a non-linear function (e.g., softmax) of the dot-product between the query and key vectors. The linear self-attention layer has been shown to have good in-context learning capabilities (Garg et al., 2022) and it exhibits a simple and interpretable structure that allows us to analyze the in-context learning process in detail (Von Oswald et al., 2023). With a slight abuse of notation, we will also use $\boldsymbol{f}_{\text{LSA}}(\boldsymbol{\theta}, \cdot)$ to denote multiple layers of linear self-attention, where the output of the previous layer is used as input for the next layer.

**Tokenization and pre-training.** Following the standard practice described of Von Oswald et al. (2023); Akyurek et al. (2023), we tokenize the context $\mathcal{D}_n^{(\tau)}$ with the following format:

$$\boldsymbol{E}(\mathcal{D}_n^{(\tau)}, \boldsymbol{x}_{n+1}^{(\tau)}, z) = \begin{bmatrix} \boldsymbol{x}_1^{(\tau)} & \boldsymbol{x}_2^{(\tau)} & \ldots & \boldsymbol{x}_n^{(\tau)} & \boldsymbol{x}_{n+1}^{(\tau)} \\ y_1^{(\tau)} & y_2^{(\tau)} & \ldots & y_n^{(\tau)} & z \end{bmatrix} \in \mathbb{R}^{(d+1) \times (n+1)}. \tag{2}$$

where $\boldsymbol{x}_i^{(\tau)}$ and $y_i^{(\tau)}$ are the input features and labels for task $\tau$. Note that we also include the new input $\boldsymbol{x}_{n+1}^{(\tau)}$ in the tokenization, but we mask ($z = 0$) the corresponding label as the quantity we aim to predict. Indeed, the ICL objective is to predict the label $y_{n+1}^{(\tau)}$ given the context $\{(\boldsymbol{x}_i^{(\tau)}, y_i^{(\tau)})\}_{i=1}^n$ and the new input $\boldsymbol{x}_{n+1}^{(\tau)}$. To achieve this, we need to pre-train a model using sequences of the form (2) for a diverse set of tasks $\tau$. These sequences are randomly generated using the following procedure:

$$f^{(\tau)} \sim p(f), \quad \boldsymbol{x}_i^{(\tau)} \sim p(\boldsymbol{x}), \quad \varepsilon_i \sim p(\varepsilon) \tag{3}$$

where $p(f)$, $p(\boldsymbol{x})$, and $p(\varepsilon)$ are the distributions of the true functions, input features, and noise terms, respectively. For the moment, we focus on linear regression tasks, where $f^{(\tau)}(\boldsymbol{x}) = \boldsymbol{x}^\top \boldsymbol{w}^{(\tau)}$. Consequently, we define the true function $f^{(\tau)}$ by sampling the weight vector $\boldsymbol{w}^{(\tau)}$ from a given $p(\boldsymbol{w})$. In practice, we define $p(\boldsymbol{x}) = \mathcal{U}(-a, a)^d$, $p(\varepsilon) = \mathcal{N}(0, \sigma_n^2)$, and $p(\boldsymbol{w}) = \mathcal{N}(\boldsymbol{0}, \sigma^2 \boldsymbol{I})$, where $a$, $\sigma_n^2$, and $\sigma^2$ are all given hyper-parameters. During pre-training, we optimize the model parameters $\boldsymbol{\theta}$ to minimize the mean squared error between the $n{+}1$-th predicted label and the true label $y_{n+1}^{(\tau)}$:

$$\mathcal{L}_{\text{pre-train}}(\boldsymbol{\theta}) = \mathbb{E}_{\boldsymbol{w}^{(\tau)}, \{\boldsymbol{x}_i^{(\tau)}, \varepsilon_i^{(\tau)}\}_{i=1}^{n+1}} \left( \boldsymbol{f}_{\text{LSA}}(\boldsymbol{\theta}, \boldsymbol{E}^{(\tau)})_{[d+1, n+1]} - y_{n+1}^{(\tau)} \right)^2. \tag{4}$$

where $\boldsymbol{E}^{(\tau)} = \boldsymbol{E}(\mathcal{D}_n^{(\tau)}, \boldsymbol{x}_{n+1}^{(\tau)}, 0)$ is the tokenized sequence for task $\tau$. In practice, we optimize the model parameters $\boldsymbol{\theta}$ via Eq. (4) by approximating the expectation with a batch of task sequences sampled from Eq. (3) and using stochastic gradient descent (Kingma and Ba, 2015). The whole pre-training process is illustrated in Fig. 1.

## 2.2. A brief overview of conformal prediction

The general goal of *conformal prediction* is to have a model based on observed $\mathcal{D}_n$ such that given a new feature vector $\boldsymbol{x}_{n+1} \in \mathbb{R}^d$, we can build a $100(1 - \alpha)\%$ confidence interval for the corresponding label $y_{n+1}$, where $\alpha \in (0, 1)$ is a user-defined significance level. The conformal prediction set for $y_{n+1}$ is defined as the set of values $z \in \mathbb{R}$ such that the label $y_{n+1}$ belongs to the set with probability at least $1 - \alpha$; in the literature, this is also referred to as the *typical set* (Mackay, 2003-06). For the classic linear regression task, the typicalness of $z$ can be defined based on the residuals of a linear model trained on the augmented dataset $\mathcal{D}_{n+1}(z) = \mathcal{D}_n \cup \{(\boldsymbol{x}_{n+1}, z)\}$. Under few assumptions, like exchangeability of the residuals, the conformal prediction set built in this way has good theoretical properties (e.g. the marginal coverage guarantee in Theorem 1, Appendix A.2). Despite these theoretical guarantees, the exact computation of the conformal prediction set is often infeasible in practice, since it requires to train a model for infinitely many augmented datasets $\mathcal{D}_{n+1}(z)$, for each $x_{n+1}$ in the test/validation set. To overcome this issue, several methods have been proposed in the literature (Chen et al., 2017, 2016; Fong and Holmes, 2021). Here, we will show how to use in-context learning to build conformal prediction sets in a computationally efficient way.

For the classic conformal prediction we can consider the setup of regularized empirical risk minimization as follows:

$$\widehat{\boldsymbol{w}} = \arg\min_{\boldsymbol{w} \in \mathbb{R}^d} \mathcal{L}(\boldsymbol{w}) \overset{\text{def}}{=} \sum_{i=1}^n \ell(y_i, f(\boldsymbol{w}, \boldsymbol{x}_i)) + \lambda \|\boldsymbol{w}\|_2^2, \tag{5}$$

where $\mathcal{L}(\boldsymbol{w})$ is the empirical risk, $\ell(y, \hat{y})$ is the loss function, $\lambda$ is the regularization coefficient, and $f(\boldsymbol{w}, \boldsymbol{x})$ is the linear model $f(\boldsymbol{w}, \boldsymbol{x}) = \boldsymbol{w}^\top \boldsymbol{x}$. While various loss functions can be considered, in this work we focus on the squared loss $\ell(y, \hat{y}) = (y - \hat{y})^2$.

For a new feature vector $\boldsymbol{x}_{n+1}$ and a given confidence level $\alpha \in (0, 1)$, our goal is to build a conformal prediction set $\Gamma_\alpha(\boldsymbol{x}_{n+1})$ such that

$$\mathbb{P}(y_{n+1} \in \Gamma_\alpha(\boldsymbol{x}_{n+1}) \mid \boldsymbol{x}_{n+1}) \geq 1 - \alpha. \tag{6}$$

To build the conformal prediction set, we need to define an augmented dataset $\mathcal{D}_{n+1}(z) = \mathcal{D}_n \cup \{(\boldsymbol{x}_{n+1}, z)\}$ and a new optimization problem:

$$\widehat{\boldsymbol{w}}(z) = \underset{\boldsymbol{w} \in \mathbb{R}^d}{\arg\min} \, \mathcal{L}(\boldsymbol{w}, z) \overset{\text{def}}{=} \sum_{i=1}^{n} \ell(y_i, f(\boldsymbol{w}, \boldsymbol{x}_i)) + \ell(z, f(\boldsymbol{w}, \boldsymbol{x}_{n+1})) + \lambda \|\boldsymbol{w}\|_2^2, \qquad (7)$$

where $\mathcal{L}(\boldsymbol{w}, \boldsymbol{z})$ is the augmented empirical risk. Now, for any $z \in \mathbb{R}$, we define a conformity score for $\mathcal{D}_{n+1}(z)$ as

$$
\begin{aligned}
\widehat{R}_i(z) &= \phi(y_i, f(\widehat{\boldsymbol{w}}(z), \boldsymbol{x}_i)), \quad i = \{1, \ldots, n\}, \\
\widehat{R}_{n+1}(z) &= \phi(z, f(\widehat{\boldsymbol{w}}(z), \boldsymbol{x}_{n+1})),
\end{aligned}
\qquad (8)
$$

where $\phi(\cdot, \cdot)$ is a permutation-invariant scoring function. In regression tasks, a common choice for $\phi$ is the absolute residual, i.e., $\phi(y, \hat{y}) = |y - \hat{y}|$.

We can define the *typicalness* of $z$ as

$$\widehat{\pi}(z) = 1 - \frac{1}{n+1} \operatorname{rank}(\widehat{R}_{n+1}(z)), \qquad (9)$$

where $\operatorname{rank}(\widehat{R}_{n+1}(z))$ is defined as $\sum_{i=1}^{n+1} \mathbb{1}(\widehat{R}_i(z) \le \widehat{R}_{n+1}(z))$.

If the conformity scores $\widehat{R}_i(z)$ are exchangeable and identically distributed, then the conformal prediction set with the desired guarantee in Eq. (6) can be defined as

$$\Gamma_\alpha(\boldsymbol{x}_{n+1}) = \{z \in \mathbb{R} \mid \widehat{\pi}(z) \ge \alpha\}. \qquad (10)$$

The set $\Gamma_\alpha(\boldsymbol{x}_{n+1})$ collects all the values $z$ such that $\widehat{\pi}(z)$ is greater than or equal to $\alpha$, which means that $\widehat{R}_{n+1}(z)$ is ranked lower than $(n+1)(1-\alpha)$ among all the conformity scores $\widehat{R}_i(z)$ for $i = \{1, \ldots, n\}$. This result is formally stated and proved in Vovk et al. (2005) and reported in Appendices A.1 and A.2. Generally, computing such a set is infeasible, since it requires to train a model for infinitely many augmented datasets $\mathcal{D}_{n+1}(z)$, in order to compute the conformity scores $\widehat{R}_i(z) = |y_i - f(\widehat{\boldsymbol{w}}(z), \boldsymbol{x}_i)|$ for all $i = \{1, \ldots, n\}$. In practice, one can approximate the computation of the conformity scores by training a model on a finite set of augmented datasets $\mathcal{D}_{n+1}(z)$, for a finite set of $z$ values (Angelopoulos and Bates, 2021). While this approach is computationally feasible, it requires to solve the optimization problem (7) for each $z$ value, which can be computationally expensive.

### 2.3. Building the bridge between in-context learning and conformal prediction

Now we show how to use the in-context learning capabilities of transformer models to build conformal prediction sets in a computationally efficient way. The key idea is to use the pre-trained model to predict the label $y_{n+1}$ and compute the conformity scores $\widehat{R}_i(z)$ for $i = \{1, \ldots, n\}$ and $\widehat{R}_{n+1}(z)$ for a finite set of $z$ values. This is supported by recent theoretical results on the convergence properties of transformer models and ICL. Indeed, Von Oswald et al. (2023) showed that, in case of noiseless targets, the linear self-attention layers can be seen as gradient descent steps on a loss function equivalent to Eq. (5), with $\lambda = 0$. Conversely, in the presence of noise, the transformer model has shown to converge to the Bayes-optimal solution for the linear regression task, with $\lambda = \sigma^2/\sigma_n^2$ (Garg et al., 2022). In

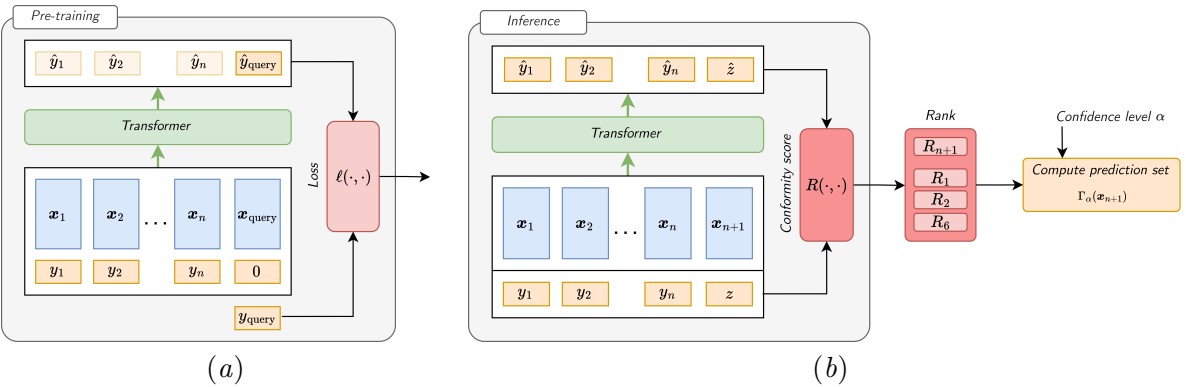

*Figure 1:* **In-context learning with conformal prediction.** The pre-training process consists of generating task sequences (Eq. (2)) and optimizing the model parameters $\boldsymbol{\theta}$ to minimize the mean squared error between the predicted and true labels (Eq. (4)). During inference, we use the pre-trained model to predict the label $y_{n+1}^{(\tau)}$ and compute the prediction interval using conformal prediction.

practice, this means that given a pre-training the model as described in Eq. (4), for a given dataset $\mathcal{D}_n$ and a new input $\boldsymbol{x}_{n+1}$, the prediction from the transformer model for the label $\widehat{y}_{n+1} = \boldsymbol{f}_{\mathrm{LSA}}(\boldsymbol{\theta}, \boldsymbol{E}(\mathcal{D}_n, \boldsymbol{x}_{n+1}, 0))_{[d+1,n+1]}$ converges to the Bayes-optimal solution for the linear regression task $\widehat{y}_{n+1} = \widehat{\boldsymbol{w}}^\top \boldsymbol{x}_{n+1}$, where $\widehat{\boldsymbol{w}}$ is the solution of Eq. (5) with $\lambda = \sigma^2/\sigma_n^2$. This is a key property that we want to exploit in our methodology.

First of all, we observe that $\widehat{\boldsymbol{w}}(z)$ in Eq. (7) is only needed to compute the conformity scores $\widehat{R}_i(z)$ and $\widehat{R}_{n+1}(z)$ in Eq. (8) via the predictions of the linear model $f(\widehat{\boldsymbol{w}}(z), \boldsymbol{x}_i)$ and $f(\widehat{\boldsymbol{w}}(z), \boldsymbol{x}_{n+1})$. Given the convergence properties of the transformer model, we can approximate the conformity scores by using the predictions of the pre-trained model. In particular, we can define the conformity scores as

$$
\begin{aligned}
\widehat{R}_i(z) &= \left| y_i - \widehat{\boldsymbol{y}}(z)_{[i]} \right|, \quad i = \{1, \dots, n\}, \\
\widehat{R}_{n+1}(z) &= \left| z - \widehat{\boldsymbol{y}}(z)_{[n+1]} \right|,
\end{aligned}
\tag{11}
$$

where $\widehat{\boldsymbol{y}}(z) = \boldsymbol{f}_{\mathrm{LSA}}(\boldsymbol{\theta}, \boldsymbol{E}_{n+1}(z))_{[d+1,:]}$ is the vector of predicted labels for the input $\boldsymbol{E}_{n+1}(z) = \boldsymbol{E}(\mathcal{D}_n, \boldsymbol{x}_{n+1}, z)$ with the context $\mathcal{D}_n$ augmented by $\{\boldsymbol{x}_{n+1}, z\}$. During pre-training, the model learns to predict $y_{n+1}$ from the masked token. At inference, different values of $z$ are used in the conformal prediction method to form confidence intervals. Since the transformer is trained to predict $y_{n+1}$, the same forward pass also produces predictions for any $y_i$. As a result, the model can predict the label $z$ for any given $z$, and also all the labels $y_i$ in $\mathcal{D}_n$, which is exactly what we need to compute the conformity scores.

The conformity scores from Eq. (11) are now based on the in-context predictions of the pre-trained model, rather than on the residuals from a linear model trained on the expanded dataset $\mathcal{D}_{n+1}(z)$. This allows computation of the conformity scores for a finite set of $z$ values with a single forward pass of the transformer, instead of solving the optimization problem Eq. (7) for each $z$. With the scores $\widehat{R}_i(z)$ and $\widehat{R}_{n+1}(z)$, it is possible to evaluate the typicalness of $z$ as defined in Eq. (9) and construct the conformal prediction set $\Gamma_\alpha(\boldsymbol{x}_{n+1})$. This process is shown in Fig. 1(b) and described in Algorithm 1. Appendix A.3 provides a

---

**Algorithm 1:** Conformal prediction with in-context learning

---

**Input:** Pre-trained model $\boldsymbol{f}_{\mathrm{LSA}}(\boldsymbol{\theta}, \cdot)$, context $\mathcal{D}_n$, test inputs $\boldsymbol{X}_{\mathrm{new}}$, confidence level $\alpha$, grid of $z$ values $\mathcal{Z}$

**Output:** Conformal prediction set $\Gamma_\alpha(\boldsymbol{x}_{n+1})$ for each $\boldsymbol{x}_{n+1} \in \boldsymbol{X}_{\mathrm{new}}$

**for** $\boldsymbol{x}_{n+1} \in \boldsymbol{X}_{\mathrm{new}}$ **do**

   **for** $z \in \mathcal{Z}$ **do**

      $\boldsymbol{E}_{n+1}(z) = \boldsymbol{E}(\mathcal{D}_n, \boldsymbol{x}_{n+1}, z)$ ;          // Tokenize the augmented dataset

      $\widehat{\boldsymbol{y}}(z) = \boldsymbol{f}_{\mathrm{LSA}}(\boldsymbol{\theta}, \boldsymbol{E}_{n+1}(z))_{[d+1,:]}$ ;          // Predict the labels

      $\widehat{R}_i(z) = \left| y_i - \widehat{\boldsymbol{y}}(z)_{[i]} \right|$ for $i = \{1, \ldots, n\}$ ;          // Compute conformity scores

      $\widehat{R}_{n+1}(z) = \left| z - \widehat{\boldsymbol{y}}(z)_{[n+1]} \right|$

      $\widehat{\pi}(z) = 1 - \frac{1}{n+1} \operatorname{rank}(\widehat{R}_{n+1}(z))$ ;          // Compute typicalness

   **end**

   $\Gamma_\alpha(\boldsymbol{x}_{n+1}) = \{z \in \mathcal{Z} \mid \widehat{\pi}(z) \geq \alpha\}$ ;          // Build the conformal prediction set

**end**

---

theoretical justification for the use of the in-context predictions by extending the coverage guarantee of the conformal prediction framework to the case of ICL.

**Practical considerations.** A couple of practical considerations are in order. First, we need to choose a finite set of $z$ values to compute the conformity scores. In our case, we consider a grid of $z$ values $\mathcal{Z}$, chosen based on the range of the labels in the training set. In particular we consider a $K$-sized grid s.t. $z \in [y_{\min} - 0.25\Delta, y_{\max} + 0.25\Delta]$, where $y_{\min}$ and $y_{\max}$ are the minimum and maximum labels in the training set, and $\Delta = y_{\max} - y_{\min}$. This choice allows us to cover the range of the labels in the training set and to have a good resolution for the conformity scores (for more details, see Remark 5 in Lei (2017) and Appendix A.4). Second, we observe that the two for-loops in Algorithm 1 can be parallelized, since the computation of the conformity scores for different $z$ values and different input features $\boldsymbol{x}_{n+1}$ are independent. This allows us to compute the conformal prediction set for multiple input features in parallel, which can be useful in practice to speed up the computation. In our experiments, we use JAX's vectorization to parallelize the computation of the conformity scores and the typicalness for multiple input features, but other parallelization strategies (e.g. explicitly building a batch with all combination of $z$ and $\boldsymbol{x}_{n+1}$) can be used as well.

## 3. Experiments

In this section, we will describe the experiments we conducted to evaluate the quality of the predictive intervals produced by the proposed method.

### 3.1. Experimental setup

With the following experiments, we aim to answer the research question: "*How good are the predictive intervals produced by combining in-context learning with conformal prediction?*" To answer this, we first need to define what we will use to compare the quality of the uncertainty estimates produced by our method. Given the theoretical and empirical equivalence between ICL and ridge regression (van Wieringen, 2015), we can leverage the mechanistic interpretation of ICL to build exact oracle conformal predictors. This oracle is computed using ridge regression models on the same data used in the context of the transformer model, and then

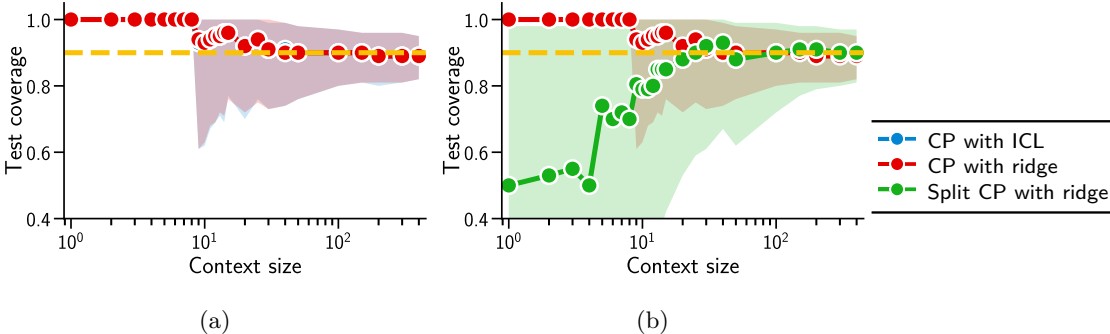

(a)                                                                 (b)

*Figure 2:* **Test coverage as a function of context size.** In (a) both *CP with ICL* and *CP with ridge* converge to the theoretical value of $1 - \alpha = 0.90$ ( ▬▬ ) as the context size increases, and the behavior of the two methods is similar. The comparison with *split CP with ridge* (b) shows that the latter has higher variance for smaller context sizes.

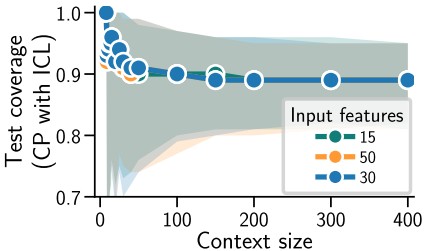

*Figure 3:* **Coverage of *CP with ICL* across varying input dimensions (15, 30, and 50).** Regardless of the input dimension, the test coverage converges to the theoretical value as the context size increases.

*Table 1:* **Compute time for different methods and context sizes.** The computation time is reported in seconds. *CP with ICL* is comparable, when not faster, than the two other methods.

| Context | CP with ICL | CP with ridge | Split CP with ridge |
|---|---|---|---|
| 50 | 0.31 (0.29, 0.35) | 0.41 (0.38, 0.46) | 0.40 (0.36, 0.45) |
| 100 | 0.32 (0.29, 0.41) | 0.42 (0.40, 0.49) | 0.41 (0.37, 0.49) |
| 300 | 0.38 (0.33, 0.53) | 0.42 (0.39, 0.50) | 0.43 (0.38, 0.60) |

following the conformal prediction algorithm to produce the predictive intervals. We will use this oracle (which we refer to as *CP with ridge*) as the baseline. We will also compare against split conformal prediction (Angelopoulos and Bates, 2021), which we will refer to as *split CP with ridge*: it uses the same ridge regression model as the oracle but builds the conformal predictor on a separate validation set instead of the augmented dataset. Appendix A.5 provides more details on the split conformal prediction method. Note that we are not interested in comparing other methods for uncertainty quantification for linear regression, such as Bayesian inference or ensemble methods (Prado et al., 2021; Lakshminarayanan et al., 2017): the scope is solely to evaluate the quality of the predictive intervals produced by a transformer model with in-context learning against the oracle conformal predictor. Finally, we evaluate our method described in § 2 the *CP with ICL*, which uses the transformer model with in-context learning to produce the predictive intervals. If not otherwise specified, we report all results as median and 95% confidence intervals over 1000 test runs.

### 3.2. Evaluating the quality of the predictive intervals

We start by analyzing the test coverage, which is the proportion of the test set that falls within the predictive intervals, across different context sizes. The test coverage is computed on a separate test set, which is not used during the training of the models, and it is collected for 1000 independent runs (each run is a set of $n$ training/in-context points and 100 testing points). In Fig. 2(a), we see that the behavior of *CP with ICL* is very similar to the *CP with ridge* oracle, with both methods converging to the theoretical value of $1 - \alpha = 0.90$ as the context size increases. Differently, *split CP with ridge* in Fig. 2(b) shows higher variance and less reliable prediction intervals, particularly for smaller context sizes. This is expected, as the split conformal prediction method is known to be a biased estimator of the conformal set (Angelopoulos and Bates, 2021). We also analyze the test coverage by varying the input feature dimensions in Fig. 3, where we observe that the test coverage converges well to the theoretical value regardless of the input dimension. Finally, in Table 1 we show the computation time of the different methods as a function of the context size. Note that all methods are fairly compared, as we made sure to enable all possible optimizations in the code (e.g. vectorization and just-in-time compilation) and to run the experiments on the same hardware. From these results, we can conclude that *CP with ICL* exhibits similar performance to the oracle method *CP with ridge*, while being as computationally efficient as the other methods. This suggests that *CP with ICL* can provide a good trade-off between computational efficiency and quality of the predictive intervals.

Next, we analyze the Wasserstein distance between the typicalness values (as a proxy of the predictive distribution) of *CP with ICL* and *CP with ridge* as a function of the ratio between the input dimension and the context size. The Wasserstein distance is a measure of the discrepancy between two probability distributions, and in this case, it quantifies the difference between the predictive distributions produced by the two methods. This metric is a stronger measure of the quality of each method, as it captures the differences in the entire predictive distribution, rather than just the coverage (additional details in Appendix C.1).

In Fig. 4, we observe behavior that differs from the test coverage: the Wasserstein distance does not decrease monotonically with the context size, with a peak at $n = d$. This phenomenon may be attributed to either overfitting noise in the labels by the ICL method (when $n$ is large on the left hand side of the curve) or underfitting of the data by the ridge regression method (when $n$ is small on the right hand side of the curve). We report this result to highlight non-trivial aspects of the quality of the predictive intervals produced by the ICL method, and we plan to further investigate this behavior in future work. This result suggests the presence of a "double descent" phenomenon,

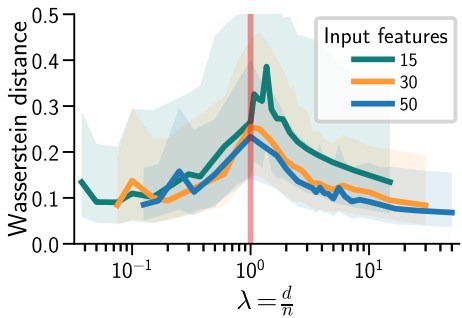

*Figure 4:* **Wasserstein distance between predictive distributions as a function of** $d/n$**.** Regardless of the input dimension $d$, the Wasserstein distance does not decrease monotonically with the data features ratio, with a peak at $d/n = 1$.

similar to observations in prior research (Belkin et al., 2019; Nakkiran et al., 2020). In

any case, this result suggests that the *CP with ICL* method is able to provide a good approximation of the ridge regression method, even in small-sample scenarios.

### 3.3. Inference on a different distribution from pre-training

In this section, we evaluate the performance of the quality of the predictions obtained by the *CP with ICL* method when the data is drawn from a different distribution than the one used during pre-training. Recall Eq. (3) where we define the data generation process with $p(\boldsymbol{w})$ and $p(\boldsymbol{x})$ as the prior distributions of the linear weights and the input data, respectively. Explicitly, we refer to these distributions as $p_{\mathrm{pt}}(\boldsymbol{w})$ and $p_{\mathrm{pt}}(\boldsymbol{x})$, to indicate that they are the distributions used during the pre-training of the transformer model. During inference, both the input data and the linear weights are drawn from a potentially different distribution, which we refer to as $p_{\mathrm{inf}}(\boldsymbol{w})$ and $p_{\mathrm{inf}}(\boldsymbol{x})$. Note that this is a different setting from the classical out-of-distribution and covariate shift problems. In our case, the training and test data used to build the in-context points are still drawn from the same distribution, which is different from the distribution used to pre-train the transformer model (Garg et al., 2022). The objective of this experiment is to evaluate the robustness of the *CP with ICL* method to distribution shifts in the input data and latent function parameters during inference. In particular, we set $p_{\mathrm{inf}}(\boldsymbol{w}) = \mathcal{N}(\boldsymbol{0}, \sigma^2 \boldsymbol{I})$ and $p_{\mathrm{inf}}(\boldsymbol{x}) = \mathcal{U}(-a, a)^d$ and we analyze the performance of the *CP with ICL* method as a function of the input range and the weight scale. The results are summarized in Fig. 5. We start by analyzing the coverage at different input ranges and weight scales in Fig. 5($a$) and Fig. 5($b$), respectively. We observe that the coverage is robust to changes in the input range and the weight scale, with the method providing reliable prediction intervals regardless of the covariate shift. Next, we analyze the Wasserstein distance between the predictive distributions of the *CP with ICL* method and the *CP with ridge* oracle in Fig. 5($c$) and Fig. 5($d$), respectively. Here, we observe a different behavior: (1) as expected, we have a minimum Wasserstein distance at the same input range and weight scale used during pre-training ($a = 1$ and $\sigma = 1$), and (2) while the Wasserstein distance increases as the input range and the weight scale deviate from the pre-training distribution, the trend is not monotonic.

### 3.4. Scaling laws for conformal prediction with in-context learning

We also study the quality of the conformal prediction intervals as a function of the compute required to pre-train the transformer model. Akin to the analysis by, e.g., Kaplan et al. (2020); Hoffmann et al. (2022), we investigate the scaling laws in a compute-constrained setting, which we intend to use to inform the design of future experiments. Following the methodology of Hoffmann et al. (2022), we parameterize the loss as a function of the number of model parameters $N$ and the number of training data points $D$ (alternatively, the number of training steps $T$ can be used as a proxy for $D$). Note that the total compute required to train the model is a deterministic (but unknown) function of $N$ and $D$.

We decompose the loss $\mathcal{L}$ as a function of three distinct terms as follows:

$$f(N, D \mid \alpha, \beta, A, B, E) = E + \frac{A}{N^\alpha} + \frac{B}{D^\beta} \,. \tag{12}$$

While originally Eq. (12) is used to model the perplexity of a language model, we adapt it to model the quality of the predictive intervals produced by the conformal prediction algorithm

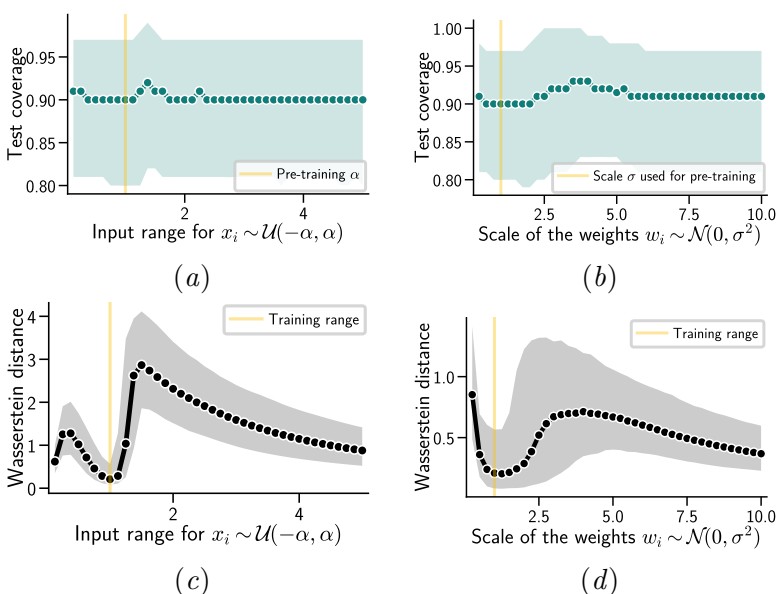

*Figure 5:* **Inference on a different distribution from pre-training.** In (a) and (b) we show the test coverage as a function of the input range and the weight scale, respectively. In (c) and (d) we show the Wasserstein distance between the predictive distributions as a function of the input range and the weight scale, respectively.

(see § 3.1). Nonetheless, we also assume that the quality of the predictive intervals is a function of the model's capacity and the amount of data used to train it.

In our experiment, we aim to estimate the parameters $\alpha$, $\beta$, $A$, $B$, and $E$ in Eq. (12) using a set of models trained with different sizes and compute budgets. We then use the estimated scaling laws to predict the optimal model size and data points for a given compute budget. The results are summarized in Fig. 6.

We now have an estimate of the scaling laws which we can use to predict the best allocation parameters/data for a given compute budget $C$:

$$\widehat{N}, \widehat{D} = \underset{N,D}{\arg\min} f(N, D \,|\, \alpha, \beta, A, B, E) \qquad \text{s.t.} \quad \text{FLOPs}(N, D) = C \,, \tag{13}$$

where $\text{FLOPs}(N, D)$ is the number of floating-point operations required to train the model with $N$ parameters on $D$ tokens.

The solution of Eq. (13) gives us a power law relationship, with the optimal number of parameters and data points scaling as $N \propto C^a$ and $D \propto C^b$ (with $a = \frac{\beta}{\alpha+\beta}$ and $b = \frac{\alpha}{\alpha+\beta}$). We refer to the results of Hoffmann et al. (2022) for the exact solution of Eq. (13). In our experiment we estimated $a = 0.62$ and $b = 0.38$. This suggests that the optimal model size scales faster than the amount of data, which slightly contradicts the trends observed of Hoffmann et al. (2022).

We hypothesize that this discrepancy is due to the fact that we are evaluating the quality of the predictive intervals of conformal prediction, rather than the perplexity of a language model. This suggests that for uncertainty quantification tasks, models with larger capacity are more beneficial than models trained on more data.

This might be connected to a recent phenomenon known as "uncertainty collapse" (Kirsch, 2024; Fellaji and Pennerath, 2024), where large models are able to capture the epistemic uncertainty better than smaller models. We plan to further investigate this behavior in future work.

Finally, we use the estimated scaling laws to predict the optimal model size and data points for a given compute budget. In Fig. 6, we start by showing the predicted performance for compute-efficient frontier (i.e., the best allocation for each compute budget). Then, we consider a compute budget of $C = 3e12$ and $C = 1e13$ FLOPs, we predict the optimal model size and we train two models with the predicted allocation. As shown in Fig. 6, the performance of these two models are close to the predicted compute-efficient frontier, which suggests that the such scaling laws are a good proxy for the quality of the predictive intervals produced by the conformal prediction algorithm.

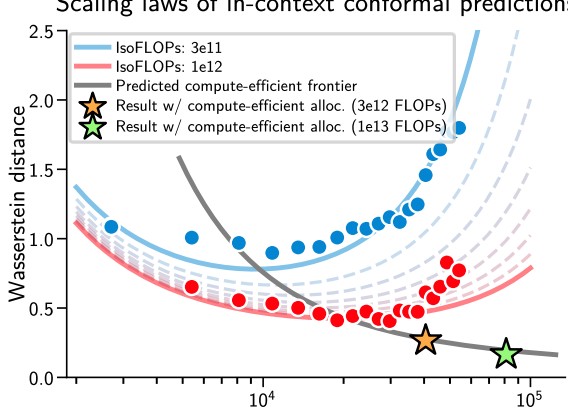

*Figure 6:* **Scaling laws for conformal prediction with in-context learning.** We fit and record the performance of 40 models with $3e11$ (●) and $1e12$ (●) FLOPs budgets. After fitting the scaling law, we draw the two isoFLOPs contours for $C = 3e11$ (▬) and $C = 1e12$ (▬), and the predicted compute-efficient frontier (▬). Finally we show the performance of two compute-efficient models trained with the best allocation of $C = 3e12$ (⭐) and $C = 1e13$ (⭐) . Both models are close to the predicted frontier.

## 4. Conclusions

We proposed an uncertainty estimation method for transformer models, exploiting in-context learning and conformal prediction to produce coverage-guaranteed intervals with a single forward pass. Building on mechanistic insights of in-context learning, we designed *CP with ICL*, maintaining the theoretical guarantees of exact conformal methods. Our empirical analysis benchmarks *CP with ICL* against oracle conformal predictors, enabling rigorous evaluation of uncertainty estimates without distributional or parametric assumptions, and without approximations. Experiments on synthetic data confirm that our method achieves reliable coverage with superior computational efficiency over classical approaches.

**Limitations and Future Work.** Our method is currently limited to synthetic datasets, which do not fully reflect real-world complexity. Extending to more complex data will allow broader evaluation. The analysis so far focuses on regression tasks; extending to classification will move us closer to token-level uncertainty estimation in language models. Also, we use non-autoregressive models, unlike in the case of real-world language models. The current implementation relies on a simplified version of the transformer with linear self-attention without causal mask, as this doesn't break the exchangeability assumption required by standard conformal prediction. However, recent works have shown how full conformal prediction algorithms can be extended to nonexchangeable data and asymmetric predictive models (Barber et al., 2023), which can be used to extend our approach to autoregressive models.

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

## Appendix A. Full Conformal Prediction and Coverage Guarantees

### A.1. A Primer on Full Conformal Prediction

Conformal prediction is a distribution-free framework for constructing valid prediction intervals or sets with guaranteed finite-sample coverage. Formally, let $\mathcal{D}_n = \{(\boldsymbol{x}_1, y_1), \ldots, (\boldsymbol{x}_n, y_n)\}$ denote an observed dataset where $\boldsymbol{x}_i \in \mathbb{R}^d$ are features and $y_i \in \mathbb{R}$ are the outputs. The objective is to build a prediction set $\Gamma_\alpha(\boldsymbol{x}_{n+1}) \subset \mathbb{R}$ for a new feature vector $\boldsymbol{x}_{n+1}$, such that:

$$\mathbb{P}(y_{n+1} \in \Gamma_\alpha(x_{n+1})) \geq 1 - \alpha \tag{14}$$

where $\alpha \in (0, 1)$ is the user-defined significance level. Conformal prediction relies on conformity scores, which quantify how well a candidate label $z \in \mathbb{R}$ conforms to the training data distribution. For regression tasks, a common choice is the absolute residual:

$$R_i(z) = |y_i - f(\widehat{\boldsymbol{w}}(z), \boldsymbol{x}_i)|, \quad i = 1, \ldots, n \tag{15}$$

where $f(\widehat{\boldsymbol{w}}(z), \boldsymbol{x}_i)$ is the prediction of a model trained on the augmented dataset $\mathcal{D}_{n+1}(z) = \mathcal{D}_n \cup \{(\boldsymbol{x}_{n+1}, z)\}$ and $\widehat{\boldsymbol{w}}(z)$ is the corresponding fitted parameter vector.

A crucial quantity in conformal prediction is the *typicalness* function:

$$\pi(z) = 1 - \frac{1}{n+1} \sum_{i=1}^{n+1} \mathbb{I}\left(R_i(z) \leq R_{n+1}(z)\right) \tag{16}$$

Here, $R_{n+1}(z) = |z - f(\widehat{w}(z), x_{n+1})|$ is the conformity score of the test point under the candidate label $z$.

The conformal prediction set $\Gamma_\alpha(x_{n+1})$ is then defined as:

$$\Gamma_\alpha(x_{n+1}) = \{z \in \mathbb{R} \,|\, \pi(z) \geq \alpha\} \tag{17}$$

This set comprises all candidate labels whose conformity score is typical relative to the training data. The theoretical coverage guarantee of conformal prediction, under the assumption of exchangeability of the data points, follows from the distribution-free property of the rank-based construction of $\pi(z)$ (Vovk et al., 2005; Shafer and Vovk, 2008)

The full conformal prediction approach used in this paper integrates this classical framework with ICL. The ICL-enabled transformer model, pre-trained over synthetic linear regression tasks, predicts the full set of outputs $\{y_i\}_{i=1}^{n+1}$ for any input-augmented sequence $E(\mathcal{D}_n, \boldsymbol{x}_{n+1}, z)$, where $z$ acts as the candidate label for the test input. Crucially, because of the transformer's one-pass inference capability, conformity scores across all data points and all candidate $z$ values can be computed with a single forward pass.

### A.2. Coverage Guarantee of Full Conformal Prediction

We now formally state and prove the marginal coverage guarantee of the full conformal prediction method described above. The result follows from the exchangeability of the data points and standard conformal prediction theory (Vovk et al., 2005; Shafer and Vovk, 2008; Lei et al., 2018). In particular, we restate Theorem 2.1 in (Lei et al., 2018).

**Theorem 1 (Marginal Coverage Guarantee)** *Assume that the data points $\{(\boldsymbol{x}_i, y_i)\}_{i=1}^n$ and the test point $(\boldsymbol{x}_{n+1}, y_{n+1})$ are exchangeable. Consider the full conformal prediction set $\Gamma_\alpha(\boldsymbol{x}_{n+1})$ constructed as:*

$$\Gamma_\alpha(x_{n+1}) = \left\{ z \in \mathbb{R} \,\middle|\, \pi(z) = 1 - \frac{1}{n+1} \sum_{i=1}^{n+1} \mathbb{I}\left(R_i(z) \leq R_{n+1}(z)\right) \geq \alpha \right\}$$

*where the conformity scores $R_i(z)$ are computed as:*

$$R_i(z) = |y_i - \hat{y}_i(z)|, \quad R_{n+1}(z) = |z - \hat{y}_{n+1}(z)|$$

*Then, the prediction set $\Gamma_\alpha(x_{n+1})$ satisfies:*

$$\mathbb{P}\left(y_{n+1} \in \Gamma_\alpha(\boldsymbol{x}_{n+1})\right) \geq 1 - \alpha$$

*where the probability is over the joint distribution of $\{(\boldsymbol{x}_i, y_i)\}_{i=1}^{n+1}$.*

**Proof** By construction, the augmented dataset $\mathcal{D}_{n+1}(z)$ consists of $n+1$ samples, including the candidate point $(\boldsymbol{x}_{n+1}, z)$. Given the assumption of exchangeability, any permutation of the dataset $\mathcal{D}_{n+1}(z)$ has the same joint distribution. Consider the conformity scores $\{R_i(z)\}_{i=1}^{n+1}$ computed on $\mathcal{D}_{n+1}(z)$. Under exchangeability, the rank of the conformity score $R_{n+1}(z)$ among all scores $\{R_i(z)\}_{i=1}^{n+1}$ is uniformly distributed over $\{1, \ldots, n+1\}$. Therefore, for any fixed candidate label $z$, we have:

$$\mathbb{P}\left(\frac{\text{rank}\left(R_{n+1}(z)\right)}{n+1} \leq \alpha\right) \leq \alpha$$

Equivalently, this implies:

$$\mathbb{P}\left(\pi(z) \geq \alpha\right) \geq 1 - \alpha$$

Since the true label $y_{n+1}$ corresponds to some candidate value $z = y_{n+1}$, and given that our prediction set $\Gamma_\alpha(x_{n+1})$ collects all $z$ such that $\pi(z) \geq \alpha$, it follows directly that:

$$\mathbb{P}\left(y_{n+1} \in \Gamma_\alpha(\boldsymbol{x}_{n+1})\right) \geq 1 - \alpha$$

Thus, the full conformal prediction set satisfies the desired marginal coverage guarantee. ∎

**Remark 2** *This result holds for any finite sample size $n$ and any model used to compute the conformity scores, including the transformer model with in-context learning as employed in this work, provided the exchangeability assumption is satisfied.*

### A.3. Generalized Conformity Scores with Symmetric Functions

Building on classical conformal prediction, we extend the coverage guarantee to more general conformity scores, in particular to the permutation-invariant function implemented by our in-context learning transformer model. This generalization is critical in our setting, where conformity scores emerge from the joint transformer output over tokenized sequences.

We recall the setup above and state the following theorem (crf Remark 2.4 in (Lei et al., 2018)).

**Theorem 3 (Coverage Guarantee with Symmetric Conformity Score)**  *Let $\mathcal{D}_n = \{(\boldsymbol{x}_1, y_1), \ldots, (\boldsymbol{x}_n, y_n)\}$ be the observed dataset and $(\boldsymbol{x}_{n+1}, y_{n+1})$ the test point. Assume $\mathcal{D}_{n+1} = \mathcal{D}_n \cup \{(\boldsymbol{x}_{n+1}, y)\}$ is exchangeable for any candidate label $y$. Define a conformity score $R_i(y)$ as:*

$$R_i(y) = f\left((\boldsymbol{x}_1, y_1), \ldots, (\boldsymbol{x}_{i-1}, y_{i-1}), (\boldsymbol{x}_{i+1}, y_{i+1}), \ldots, (\boldsymbol{x}_{n+1}, y); (\boldsymbol{x}_i, y_i)\right)$$

*where $f$ is a measurable function symmetric in its first $n$ arguments. Then, the prediction set*

$$\Gamma_\alpha(\boldsymbol{x}_{n+1}) = \{y \in \mathbb{R} \mid \pi(y) \geq \alpha\}$$

*with*

$$\pi(y) = 1 - \frac{1}{n+1} \sum_{i=1}^{n+1} \mathbb{I}\left(R_i(y) \leq R_{n+1}(y)\right)$$

*satisfies the marginal coverage guarantee:*

$$\mathbb{P}\left(y_{n+1} \in \Gamma_\alpha(\boldsymbol{x}_{n+1})\right) \geq 1 - \alpha$$

**Proof**  The key observation is that the exchangeability of $\mathcal{D}_{n+1}$ ensures the uniform distribution of the rank of $R_{n+1}(y_{n+1})$ among $\{R_i(y_{n+1})\}_{i=1}^{n+1}$, regardless of the specific form of $f$, provided that $f$ is symmetric in its first $n$ arguments.

Since $f$ is symmetric, the induced conformity scores $\{R_i(y)\}$ are equally distributed under permutation of the first $n$ data points. As a result, the rank of $R_{n+1}(y_{n+1})$ is uniformly distributed over $\{1, \ldots, n+1\}$:

$$\mathbb{P}\left(\text{rank}\left(R_{n+1}(y_{n+1})\right) \leq k\right) = \frac{k}{n+1}$$

for any $k \in \{1, \ldots, n+1\}$. By construction of $\pi(y)$, we have:

$$\pi(y_{n+1}) = 1 - \frac{\text{rank}\left(R_{n+1}(y_{n+1})\right)}{n+1}$$

Thus,

$$\mathbb{P}\left(\pi(y_{n+1}) \geq \alpha\right) = \mathbb{P}\left(\text{rank}\left(R_{n+1}(y_{n+1})\right) \leq (1-\alpha)(n+1)\right) \geq 1 - \alpha$$

which concludes the proof. ∎

**Remark 4**  *This result shows that our transformer-based in-context learning approach, which implicitly computes conformity scores through predictions based on the permutation-invariant attention mechanism, satisfies the same marginal coverage guarantees as classical conformal methods. The symmetry condition is naturally satisfied by the self-attention layers without causal mask, as their output for token $i$ depends symmetrically on the other tokens $(\boldsymbol{x}_j, y_j)$, $j \neq i$.*

## A.4. Justification of the Grid Range for Candidate Labels

An essential practical consideration in the full conformal prediction method used in this work is the selection of the candidate grid $\mathcal{Z}$ over which the conformity scores and typicalness function $\pi(z)$ are evaluated. As exact computation of $\Gamma_\alpha(x_{n+1})$ requires searching over a continuous domain, in practice we discretize the output space and compute conformity scores over a finite grid of candidate labels:

$$\mathcal{Z} = \{z \in [y_{\min}, y_{\max}]\}$$

with $[y_{\min}, y_{\max}]$ denoting the search interval. The choice of this interval is theoretically motivated by the properties of order statistics in conformal inference. Specifically, selecting $[y_{\min}, y_{\max}]$ as the sample range of observed responses incurs a coverage loss of at most $\frac{2}{n+1}$, i.e.,

$$\mathbb{P}\left(y_{n+1} \in \left[y_{(1)}, y_{(n)}\right]\right) \geq 1 - \frac{2}{n+1}$$

where $y_{(1)}$ and $y_{(n)}$ denote the minimum and maximum observed values in the training data. Assuming the data points $\{y_i\}_{i=1}^n$ and the test point $y_{n+1}$ are i.i.d. draws from an continuous distribution $p(y)$, the empirical order statistics $\{y_{(1)}, \ldots, y_{(n)}\}$ provide a natural quantile-based partitioning of the sample space. By the properties of order statistics for i.i.d. samples, the probability that a new independent draw $y_{n+1}$ falls outside the range $\left[y_{(1)}, y_{(n)}\right]$ corresponds to the probability that $y_{n+1}$ is either smaller than $y_{(1)}$ or larger than $y_{(n)}$:

$$\mathbb{P}\left(y_{n+1} < y_{(1)}\right) + \mathbb{P}\left(y_{n+1} > y_{(n)}\right)$$

It is well known (David and Nagaraja, 2004) that, for continuous distributions, the probability that a new i.i.d. sample falls below the minimum of $n$ i.i.d. samples is given by

$$\mathbb{P}\left(y_{n+1} < y_{(1)}\right) = \int F_Y(y)^n \, dF_Y(y) = \frac{1}{n+1}$$

Similarly, the probability of exceeding the sample maximum is:

$$\mathbb{P}\left(y_{n+1} > y_{(n)}\right) = \int (1 - F_Y(y))^n \, dF_Y(y) = \frac{1}{n+1}$$

where $F_Y(y)$ is the cumulative distribution function of the output variable.

Summing these contributions, the total probability of $y_{n+1}$ lying outside the sample range is:

$$\mathbb{P}\left(y_{n+1} \notin \left[y_{(1)}, y_{(n)}\right]\right) = \frac{2}{n+1}$$

Thus, the probability that $y_{n+1}$ lies within the empirical range is:

$$\mathbb{P}\left(y_{n+1} \in \left[y_{(1)}, y_{(n)}\right]\right) = 1 - \frac{2}{n+1}$$

**Remark 5** *This result is non-asymptotic and holds exactly for any finite sample size n under the i.i.d. sampling assumption. Importantly, it does not require knowledge of the distribution $p(y)$. When applied to the construction of the candidate grid $\mathcal{Z}$ in full conformal prediction, it ensures that the grid spans a region containing the true response $y_{n+1}$ with high probability.*

To further mitigate any potential loss of coverage due to discretization or boundary effects, we enlarge the search range by a fraction of its empirical span. Following this approach, we extend the search interval symmetrically beyond the empirical minimum and maximum of the observed outputs:

$$[y_{\min}, y_{\max}] = \left[ y_{(0)} - 0.25 \cdot \Delta y, \ y_{(n)} + 0.25 \cdot \Delta y \right]$$

where $\Delta y = y_{(n)} - y_{(0)}$ is the empirical range of the output variable.

**Remark 6** *The grid design strategy employed here is directly inspired by the theoretical insights of Lei (2017), specifically Remark 5 therein, which provides finite-sample justification for extending the search interval beyond the observed range of the data. This ensures robustness of the predictive intervals to sampling variability and model approximation errors, particularly in small-sample regimes.*

### A.5. Split Conformal Prediction

In contrast to the full conformal prediction framework discussed previously, split conformal prediction (split CP) offers a computationally efficient alternative for constructing marginal prediction intervals. Rather than augmenting the training set with candidate outputs and recomputing conformity scores for each trial label, split CP partitions the data into disjoint subsets, using one for training and the other for calibration. This strategy trades some statistical efficiency for computational simplicity, which is particularly advantageous when model evaluation is expensive.

Formally, let the original dataset $\mathcal{D} = \{(\boldsymbol{x}_i, y_i)\}_{i=1}^n$ be partitioned into a proper training set $\mathcal{D}_{\text{train}}$ of size $n_{\text{train}}$, and a calibration set $\mathcal{D}_{\text{cal}}$ of size $n_{\text{cal}}$, such that $N = n_{\text{train}} + n_{\text{cal}}$. The training set $\mathcal{D}_{\text{train}}$ is used to fit a regression model $\widehat{\mu} : \mathbb{R}^d \to \mathbb{R}$. This model serves as the point predictor. Given $\widehat{\mu}$, we evaluate the conformity scores on the calibration set:

$$R_i = |y_i - \widehat{\mu}(\boldsymbol{x}_i)|, \quad \text{for } (\boldsymbol{x}_i, y_i) \in \mathcal{D}_{\text{cal}}$$

These conformity scores represent the empirical residuals of the model over the held-out calibration points. To construct the prediction interval for a new input $\boldsymbol{x}_{n+1}$, we compute the quantile threshold:

$$Q = \text{Quantile}_{1-\alpha} \left( \{R_i\}_{i=1}^{n_{\text{cal}}} \right)$$

where $\alpha \in (0, 1)$ is the miscoverage rate. The resulting prediction interval for $\boldsymbol{x}_{n+1}$ is symmetric about the point prediction $\widehat{\mu}(x_{n+1})$:

$$\Gamma_\alpha^{\text{split}}(\boldsymbol{x}_{n+1}) = [\widehat{\mu}(\boldsymbol{x}_{n+1}) - Q, \ \widehat{\mu}(\boldsymbol{x}_{n+1}) + Q]$$

Under the assumption of exchangeability between calibration points and the new test point, the split CP interval retains the desired marginal coverage property:

$$\mathbb{P} \left( y_{n+1} \in \Gamma_\alpha^{\text{split}}(\boldsymbol{x}_{n+1}) \right) \geq 1 - \alpha$$

## Appendix B. Addition Experiment: Visualization of Predictive Performance of ICL vs Ridge on Single Data Point in Full CP Algorithm

Before the comparsion experiemnts, we also do one experiment that visualizes the predictive performance of In-Context Learning (ICL) versus Ridge Regression using a synthetic dataset. This experiment focuses on comparing the prediction accuracy and uncertainty quantification between these two methods when applied to a single input point.

In detail, we generate 1000 trial values, $Y_{\text{trial}}$, covering a range that extends beyond the minimum and maximum observed test outputs. For this experiment, we use a single input feature point, $x$, and run the Full Conformal Prediction (CP) algorithm with both ICL and Ridge Regression as the learning models. The Full CP algorithm is employed to calculate the prediction intervals for both models. Specifically, the algorithm involves augmenting the training data with trial values and computing the corresponding residuals and p-values ($p(y)$) to determine the prediction intervals.

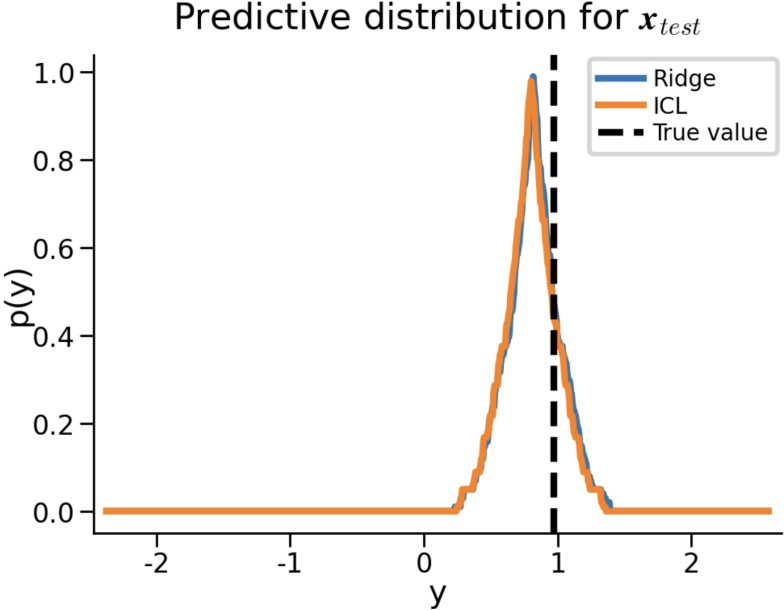

*Figure 7:* Visualization of Predictive Performance: ICL vs Ridge on Single Data Point

As shown in Figure 7, the predictive distributions for $\boldsymbol{x}_{test}$ of both models were plotted and compared against the true test values. In the figure, the x-axis represents the range of possible output values ($y$), and the y-axis represents the predictive probability ($p(y)$). The predictive distributions from both ICL and Ridge Regression models exhibit high accuracy, with peaks near the true value (indicated by the dashed line). This comparison shows that the predictive distributions from ICL closely match those from Ridge Regression, indicating that ICL performs comparably to the Ridge Regression oracle method under the Full CP framework.

## Appendix C. Experimental Setup Details

The LSA transformer model is pre-trained using masked sequences with a mean squared error (MSE) loss function. The input prompt is constructed by concatenating the input $X$ and the noisy output $y$, forming sequences that include context tokens, with the query token placed at the last position of the sequence.

During pre-training, as depicted in Fig. 1(a), the loss is computed solely on the masked query token (the last element of the sequence), aiming to minimize the difference between the predicted and actual outputs. The training process is run for 10,000 iterations, and the total FLOPs (floating-point operations) are tracked to ensure the training stays within a predefined computational budget. A FLOP cost analysis is performed at each step to log the computational requirements, and training halts if the total FLOPs exceed the set limit.

For testing, we employ conformal prediction methods to evaluate the uncertainty estimates of the model. The testing dataset consists of 1,000 test cases, where we split the dataset into training and calibration subsets, with a calibration percentage set by the configuration. We perform experiments using three methods: Split Conformal Prediction with Ridge Regression, Full Conformal Prediction with ICL, and Full Conformal Prediction with Ridge Regression. For each test case, prediction intervals (prediction bands) are computed, and their width is recorded for each method. We also measure empirical coverage, ensuring the proportion of true values falling within the prediction bands is consistent with the specified confidence level. Additionally, the computation time for each method is logged. Finally, we compute the Wasserstein distance between the p-values generated by the ICL and Ridge methods, providing a measure of distributional similarity between these approaches. Since we vary the random seeds and have 1000 test cases, the results of these tests are averaged and then stored for further analysis.

### C.1. Experimental Setup for Empirical Distribution Comparison

In the experimental section, we use the Wasserstein distance to compare the predictive behavior of two conformal prediction (CP) procedures: *CP with ridge regression* and *CP with in-context learning (ICL)*. For each method, we apply the full conformal prediction and we evaluate the empirical distribution of typicalness values $\hat{\pi}$ across a set of test points for each method and compare them via the Wasserstein-1 distance (also known as the Earth Mover's Distance). Given two probability distributions $\mu$ and $\nu$ on $\mathbb{R}$, the Wasserstein-1 distance is defined as

$$W_1(\mu, \nu) = \inf_{\gamma \in \Gamma(\mu, \nu)} \int_{\mathbb{R} \times \mathbb{R}} |x - y| \, d\gamma(x, y), \tag{18}$$

where $\Gamma(\mu, \nu)$ denotes the set of all couplings of $\mu$ and $\nu$. In one dimension, the Wasserstein-1 distance admits a closed-form expression in terms of the cumulative distribution functions (CDFs) $F_\mu$ and $F_\nu$:

$$W_1(\mu, \nu) = \int_0^1 \left| F_\mu^{-1}(t) - F_\nu^{-1}(t) \right| \, dt. \tag{19}$$

We use this formulation to compute the distance between the empirical distributions of $\hat{\pi}$ values under ridge and ICL, averaged out over the points in the test set.

## C.2. Detailed Scaling Law Methodology

We use JAX's Ahead-of-Time (AOT) compilation (Bradbury et al., 2018) to compile one training step of the model (including the gradient computation and the optimizer update) and to calculate the exact number of floating-point operations (FLOPs) required to train the model. This is more accurate than the commonly used approximation $C = 6ND$.

We fit $K = 40$ models with different sizes and compute budgets; for each training we record the loss $\mathcal{L}_i$ as well as the number of model parameters $N_i$ and the number of training tokens $D_i$. We then fit the model in Eq. (12) to the data using the `chinchilla` package (Chinchilla), by minimizing the following objective using L-BFGS (Nocedal, 1980):

$$\mathcal{L}(\alpha, \beta, A, B, E) = \frac{1}{K} \sum_{i=1}^{K} \ell_\lambda \left( \log f(N_i, D_i \,|\, \alpha, \beta, A, B, E), \log \mathcal{L}_i \right) , \tag{20}$$

where $\ell_\lambda$ is the asymmetric MAE loss function (with $\lambda = 0.1$) defined as:

$$\ell_\lambda(y, \hat{y}) = \begin{cases} y - \hat{y}, & \text{if } y - \hat{y} > 0 \\ \lambda \cdot |y - \hat{y}|, & \text{otherwise} \end{cases} \tag{21}$$

Following the suggestion in (Hoffmann et al., 2022), we fit the model starting from various initializations and select the one with the lowest loss. After fitting, we plot isoFLOPs contours of the loss function $f(N, D \,|\, \alpha, \beta, A, B, E)$, which shows the quality of the predictive intervals produced by the conformal prediction algorithm as a function of the model size and the amount of data used to train it.

