# OpenReview forum: "From predictions to confidence intervals: an empirical study of conformal prediction methods for in-context learning"
_approximateinference.org/AABI/2025/Proceedings_Track — AABI 2025 Proceedings Track_

### Official Review · Reviewer_Au7g · 2025-02-24
**Interesting idea but unclear fit**

**Rating:** 5
**Confidence:** 2

**Review:**

The paper proposes a new way to generate conformity scores by using a pre-trained model instead of repeatedly re-fitting a model. They demonstrate favorable experimental results in a simple noisy linear regression task.

I'm not sure how well-matched the paper is to the symposium. There is nothing especially Bayesian about the problem or solution and there was not a clearly articulated probabilistic model. These reservations aside, here are some general comments. I apologize in advance if these comments reveal more about my ignorance about in-context learning & conformal inference (as these are quite outside my area of expertise).


Dependence of $\hat{R}_{i}(z)$ on $z$: the notation makes it seem like the conformity score depends on $z.$ But in Algorithm 1, aren't the model parameters fixed? Or are you repeatedly pre-training the model (i.e., solving Equation 7), once for each $z$? I suspect you're not doing the latter (based on the final paragraph under Equation (11)), but it would be useful to clarify.
Relatedly, I believe a clearer presentation of the proposed algorithm would be helpful: at times you switch between notation for a linear model and a transformer and it was confusing to disentangle the two (e.g., immediately before and after Equation 11).

Sensitivity to the grid of Z values: It is strange to me to restrict the prediction set to a discrete, bounded grid, especially in a continuous outcome setting. The choice to restrict $z \in [y_{min} - 0.25\Delta, y_{max} + 0.25\Delta]$ is decidedly ad hoc. Can you provide some justification (or empirical evidence) that the results are not sensitive to this choice?

Run-time benchmarks: In Table 1, you report very similar runtimes to other conformal prediction methods. Do those other methods require repeated model re-fitting? If so, it does not seem like the proposed model offers that substantial of run-time improvements.

Comparison to probabilistic methods: I am curious how well the results compare to straightforward flexible probabilistic methods. One example is the heteroskedastic Bayesian Additive Regression Trees (pre-print available [here](https://arxiv.org/abs/1709.07542)) that asserts $y_{n} \sim \mathcal{N}(f(\mathbf{x}), \sigma^{2}(\mathbf{x})).$ One can obtain posterior predictive intervals from this model fairly easily and in my experience, the calibration is pretty good! Such a comparison would go a long way to clarifying the advantages of conformal prediction over model-based posterior prediction intervals.

Non-linear regression tasks: why did you restrict attention to linear regression tasks? Since you're pre-training an extremely flexible models, it seems natural to generate synthetic data (for pre-training) from more complicated models. Some explanation for why you limited the evaluations to linear regression (which seems like a simple-enough task to not need transformers) would be helpful.

** Update ** : Based on the authors' reply, I've increased by score to 5. Comparisons to Bayesian alternatives and using a richer set of synthetic data generating process and/or real-world benchmarks would be useful in clarifying the advantages and limitations of the proposed conformal approach.

---

### Official Review · Reviewer_yorr · 2025-03-01
**Interesting paper but with some limitations**

**Rating:** 5
**Confidence:** 4

**Review:**

The paper explores the development of an approach that uses in-context learning as a computationally efficient mechanism for constructing prediction intervals. The core idea is innovative and principled, although some theoretical statements would improve the clarity of the claims. The paper provides extensive experiments for linear regression, synthetic data and linear self-attention without a causal mask. For these settings, the experimental results prove promising.

Overall, the paper makes a useful and innovative contribution, but the claims are supported solely by empirical evidence. To support the claims and to represent an impactful research contribution, one would expect the inclusion of at least some experiments studying non-linear settings and real data.

Strengths

(1)	The paper addresses an interesting topic and is very well-written. The presentation makes it easy to follow the core ideas and the approach is clearly motivated.

(2)	The proposed method is efficient and innovative. The approach is principled.

(3)	The paper provides an interesting analysis of the scaling laws for a compute-constrained setting.

Weaknesses

(1)	Although the paper does acknowledge its limitations, they remain relatively substantial. The most important are the restrictions to linear regression, synthetic data and linear self-attention without a causal mask.

Of these, the restriction to linear regression is concerning – the model is pre-trained on linear models and then tested on linear models of the same structure. Although there are experiments where the distribution of the weights is changed, this is a very restrictive class. For the paper to make a convincing case that the proposed approach is a viable (and scalable) way to perform conformal prediction, there needs to be some experiments that extend beyond the linear setting.

The paper is primarily experimental in nature. Although there is novel algorithm development, and the principles behind the algorithm are expressed, there is no theoretical characterization of the proposed work. This being the case, one would expect a more substantial experimental exploration.

The abstract claims that the paper “bridge[s] ICL and conformal prediction, providing a theoretically grounded and practical framework for uncertainty quantification in transformer-based models.” In my view, the paper falls short of demonstrating this – there is not sufficient evidence that the proposed approach is a “practical framework”. The experimental investigation does demonstrate that the approach has potential, but given that there is only support for its utility in linear regression, there is insufficient experimental evidence for the “practical” application, where one is usually interested in the non-linear setting (as the authors illustrate, in the linear case, ridge regression provides a feasible alternative).

(2)	The conclusion states that the proposed technique has “the same theoretical guarantees”. The paper would be improved by the addition of more concrete statements about the theoretical guarantees that are provided by the proposed approach.

The paper motivates the proposed approach by arguing that the transformer model “converge[s] to the Bayes-optimal solution for the linear regression task”. The paper cited in support of this (Garg et al. 2022) is an empirical study; it does not provide theoretical support for this claim. If we assume that it does converge, then there remains the question about the nature of the convergence. A couple of propositions clarifying which guarantees hold and what the required assumptions are would improve the paper significantly.

---

### Official Review · Reviewer_SRCA · 2025-03-01

**Rating:** 7
**Confidence:** 4

**Review:**

### Summary
In this paper, the authors tackle the challenge of applying conformal prediction to in-context learning, aiming to quantify uncertainty for large language models (LLMs). Focusing on linear self-attention, they propose a method for constructing prediction intervals through a single forward pass, significantly reducing the computational overhead compared to standard conformal approaches. This efficiency stems from the observation that linear self-attention layers can be viewed as gradient descent steps on the empirical risk minimization (ERM) objective, thereby simulating the effect of retraining a model on various candidate outputs. A comprehensive empirical study confirms that the proposed method achieves reliable coverage on synthetic regression tasks, matching an oracle model's performance. Additionally, experiments under covariate shift suggest that the proposed approach remains robust to changes in data distribution.

### Strength
- The paper is well-written and logically organized, introducing key ideas in in-context learning, transformer architectures, and conformal prediction with clarity.
- It is, to the best of my knowledge, the first work to rigorously combine in-context learning with conformal prediction. The approach is novel and could have considerable impact, as uncertainty estimation is crucial for addressing issues like hallucination in LLMs.
- Building on the foundations of conformal prediction and linear transformers ensures distribution-free coverage guarantees for prediction intervals.
- The authors present a comprehensive experimental evaluation, including detailed analysis under distribution shift scenarios, thereby elucidating scaling properties relevant to uncertainty in in-context learning.

### Weakness and Questions
- One limitation is that experiments are restricted to synthetic linear regression scenarios, which may not fully capture the complexity of real-world data distributions.
- Since the in-context learning capabilities of large language models often hinge on autoregressive generation, it would be valuable for the authors to comment further on how their method might extend to or integrate with the standard, causal form of attention.

---

### Official Review · Reviewer_jTtm · 2025-03-01
**Transformers are perfectly suited for performing conformal prediction. A very well done study.**

**Rating:** 9
**Confidence:** 4

**Review:**

This paper provides a technique for uncertainty quantification for in-context learners based on transformer neural network architectures using conformal prediction. Conformal prediction provides potentially scalable methods for computing confidence intervals for the predictions of transformers. This paper is very topical at this time, as enabling UQ for the outputs of large language models provides a way to measure confidence in an LLM's outputs. While the paper only considers an idealized linear version of the transformer, the principles and results provide invaluable insight for future studies on more complex and real-world applications of the transformer.

Transformers are able to perform so-called in-context learning, in which example pairs of a particular task can be fed into a pre-trained model, thereby fine-tuning the model to perform the task with context defined by the example pairs. Conformal prediction creates confidence intervals for a given model on a new input by training multiple new models. While conformal prediction is simple to implement, it is computationally taxing as it requires training multiple models. Transformers and their in-context learning abilities are perfectly able to bypass the computationally expensive parts of conformal prediction, as the "training" of a new model is equivalent to evaluating the transformer model on new example data. The proposed methodology is simple and quite clever, showing that the properties of in-context learning fit exactly with the computational requirements of conformal prediction.

The study is excellently executed—the research question is well-scoped and well-defined, the work places itself in the context of other UQ methods, provides a concise introduction to conformal prediction, and the numerical results are informative. For these reasons, I give the top rating for this paper. The only drawback of this paper, which they mention in the conclusion, is that they only consider linear transformers, mostly because their uncertainties can be compared with baselines reliably. Perhaps it might be helpful to speculate on the potential impact and difficulties of conformal prediction for more "real-world" transformers and LLMs.

Suggestions:

To make the paper self-contained, it may be helpful to include some basic results about conformal prediction, perhaps a simple proof for why the typicalness function defines the prediction set such that (6) is true.

---

### Official Review · Reviewer_6NPG · 2025-03-03
**Creative idea and interesting paper, clarity could be improved**

**Rating:** 7
**Confidence:** 3

**Review:**

__Quality__
I found that the paper was of overall good quality.
The main text gave sufficient detail (though clarity could be improved) and the experiments seem thorough.

__Clarity__
I thought the clarity of the paper could be significantly improved.
For example, as far as I understood, the notation was not fully consistent (see “other comments” point 1).
Also, I did not fully understand how the transformer model, which seems to be pre-trained with masked z = 0 (see figure 1 (a)) is then deployed using different values of z (see figure 1 (b)).
Given that this is a central part of the algorithm, I thought it should be explained more clearly.

__Originality__
I thought the main idea of the paper was creative and interesting.
Leveraging conformal prediction to obtain confidence intervals and perform uncertainty quantification for in-context learning is, to my knowledge, a novel idea.

__Significance__
The method presented in this paper could be significant for facilitating uncertainty estimation for the in-context learning setting.

__Other comments__
1. Notation seems inconsistent. In “Notation” the authors say “Given a vector __a__, we denote it’s ith element as $\mathbf{a}_i,$ (so $\mathbf{a}_i$ is a scalar) but then a few lines later they use $\mathbf{x}_i \in \mathbb{R}^d.$ I suggest reviewing the notation to make it consistent.
2. Some explanations can be made clearer. For example, the setup defines “we mask (z = 0)”, but then the authors proceed to allow the values of $z$ to vary (e.g. algorithm 1). On this specific point, Figure 1(b) suggests the transformer can receive a non-zero (i.e. non-masked) entry for the (n+1)th point (marked z) when it has only been trained with this entry masked.
3. What are the confidence intervals shown in Figures 2, 3, 4 and 5? Are they 95% confidence intervals in the corresponding quantity shown on the y-label?

---

### Meta-Review · Area_Chair_nbHX · 2025-03-16

**Recommendation:** Accept
**Confidence:** 4

**Metareview:**

This paper provides an empirical study of whether in-context learning (ICL) in transformer models can be leveraged to efficiently perform conformal prediction. Effectively, this provides the transformer a (frequentist) coverage guarantee on its predictions.

Most reviewers agree that this work is interesting and useful in a bid to provide/improve uncertainty quantification in practically relevant models. Nevertheless, some improvement in the manuscript can be done, as some other reviewers rightly pointed out. I urge the authors to address them carefully in the final version.

---

### Decision · Program_Chairs · 2025-03-18

Accept